

# QTL mapping and candidate gene analysis of low temperature germination in rice (*Oryza sativa* L.) using a genome wide association study

Feng Mao[1,2], Depeng Wu[2,3], Fangfang Lu[2], Xin Yi[2,3], Yujuan Gu[2],
Bin Liu[2], Fuxia Liu[2,3], Tang Tang[2,3], Jianxin Shi[4], Xiangxiang Zhao[2,3],
Lei Liu[2,3] and Lilian Ji[1]

[1] School of Chemistry and Life Sciences, Suzhou University of Science and Technology, Suzhou, Jiangsu, China
[2] Jiangsu Key Laboratory for Eco-Agriculture Biotechnology around Hongze Lake, Huaiyin Normal University, Huai'an, Jiangsu, China
[3] Jiangsu Collaborative Innovation Center of Regional Modern Agriculture and Environment Protection, Huaiyin Normal University, Huai'an, Jiangsu, China
[4] School of Life Sciences and Biotechnology, Shanghai Jiao Tong University, Shanghai, China

## ABSTRACT

Low temperature germination (LTG) is a key agronomic trait in rice (*Oryza sativa* L.). However, the genetic basis of natural variation for LTG is largely unknown. Here, a genome-wide association study (GWAS) was performed using 276 accessions from the 3,000 Rice Genomes (3K-RG) project with 497 k single nucleotide polymorphisms (SNPs) to uncover potential genes for LTG in rice. In total, 37 quantitative trait loci (QTLs) from the 6th day (D6) to the 10th day (D10) were detected in the full population, overlapping with 12 previously reported QTLs for LTG. One novel QTL, namely *qLTG1-2*, was found stably on D7 in both 2019 and 2020. Based on two germination-specific transcriptome datasets, 13 seed-expressed genes were isolated within a 200 kb interval of *qLTG1-2*. Combining with haplotype analysis, a functional uncharacterized gene, *LOC_Os01g23580*, and a seed germination-associated gene, *LOC_Os01g23620* (*OsSar1a*), as promising candidate genes, both of which were significantly differentially expressed between high and low LTG accessions. Collectively, the candidate genes with favorable alleles may be useful for the future characterization of the LTG mechanism and the improvement of the LTG trait in rice breeding.

## INTRODUCTION

Rice (*Oryza sativa* L.) is an important staple food that feeds nearly half of the world (*Khush, 2005*; *Sreenivasulu, Pasion & Kohli, 2021*). Due to its tropical and subtropical origin, rice is susceptible to low temperature at all phases of growth (*Cheng et al., 2007*). A temperature of 25–35 °C is optimal for the growth of rice, and temperatures below 15 °C can cause poor seed germination and subsequently bad seedling establishment (*Fujino*

Corresponding authors
Lei Liu, leiliu_cell@163.com
Lilian Ji, jilian@usts.edu.cn

*et al., 2004*). However, more than 15 million hectares of rice cultivated worldwide are threatened by low temperatures, especially in Japan, South Korea, North Korea and Northeast China (*Song et al., 2018*). On the other hand, direct seeding has replaced conventional transplanting as it is both labor-saving and lower in cost, which requires good germination characteristics for rice seeds in low temperature, since the temperatures during the sowing period in the spring planting season are frequently below 15 °C in temperate and high-altitude regions (*Fujino et al., 2004*; *Fujino & Matsuda, 2010*; *Sales et al., 2017*; *Yang et al., 2020b*). Therefore, it is important to uncover the genetic basis of LTG and apply the findings to rice breeding in order to meet the challenges mentioned above.

In rice, LTG is a complex trait that is genetically controlled by multiple quantitative trait loci (QTLs) (*Fujino et al., 2008*). One common method used to study genetic basis is QTL analysis using bi-parental mapping populations (*Huang et al., 2010*). Generally, Japonica cultivars are more cold-tolerant than Indica cultivars (*Ma et al., 2015*). Most bi-parental populations used in QTL analysis have been derived from a cross between a cold-tolerance Japonica variety and a cold-sensitive Indica group (*Jiang et al., 2006*; *Ji et al., 2008*; *Li et al., 2013*; *Ranawake et al., 2014*; *Jiang et al., 2017*). Researchers identified five QTLs on chromosomes 2, 4, 5, and 11 in a Nipponbare × Kasalath cross (*Miura et al., 2001*). Through USSR5 and N22, 11 QTLs for LTG were unveiled on chromosomes 3, 4, 5, 9, 10 and 11 (*Jiang et al., 2006*). By crossing varieties Kinmaze and DV85, two QTLs were found located on chromosomes 7 and 11 (*Ji et al., 2008*). *Li et al. (2013)* detected three major QTLs for LTG and characterized *qLTG-9* to a region of ~72 kb which contained five potential genes explaining 12.12% of the phenotypic variation. A separate study used recombinant inbred lines from a Japonica and Indica cross and found five QTLs for LTG that explained 5.7–9.3% of the total phenotypic variance (*Ranawake et al., 2014*). *Satoh et al. (2015)* reported four QTLs responsible for LTG on chromosomes 1, 3, and 11 in a European rice variety. *Borjas, De Leon & Subudhi (2015)* found 49 QTLs related to LTG distributed on 10 chromosomes in US weedy rice. In addition, six QTLs distributed across chromosomes 1, 4, 8, and 11 were characterized for LTG by crossing Changhui 891 and 02428 (*Jiang et al., 2017*). Among the identified QTLs, only one QTL, *qLTG3-1*, has been cloned, encoding a protein with unknown molecular function that may be involved in tissue weakening (*Fujino et al., 2008*).

Compared with a bi-parental QTL analysis, a genome-wide association study (GWAS) is a more efficient way to identify the genes underlying a complex trait as it has the advantage of being able to study abundant variations in natural populations (*Huang et al., 2010*). Recently, a GWAS has been used to identify QTLs for LTG. *Fujino et al. (2015)* conducted a GWAS using 63 accessions with 117 markers and discovered 17 QTLs associated with LTG, nine of which were co-localized with QTLs identified before. Using a core collection (Rice Diversity Panel 1, RDP1) of rice, a total of 42 QTLs were identified as being associated with cold tolerance during the germination and seedling stages (*Shakiba et al., 2017*). Through a GWAS, 11 QTLs were found to be associated with LTG among Rice Diversity Panel 2 (RDP2) and two candidate genes were narrowed down

(*Yang et al., 2020c*). *OsSAP16* was cloned using 187 natural accessions by GWAS in rice (*Wang et al., 2018b*). *Yang et al. (2020b)* found 159 LTG-related QTLs in Indica accessions, only 12 of which were co-localized with previously reported cold tolerant QTLs. Consequently, a GWAS can identify new QTLs for LTG and provide new insights in to the genetic basis of LTG in rice.

In this study, a collection of 276 rice accessions from the 3K-RG project with high density SNPs were used to perform a GWAS in order to uncover potential QTLs and identify candidate genes for LTG. The favorable haplotype and SNPs affecting gene expression from two candidate genes for LTG were identified. These results provide a basis for molecular breeding to enhance LTG and further elucidate the mechanisms in rice.

## MATERIALS AND METHODS

### Plant materials

In this study, a collection of 276 rice accessions were selected from the 3K-RG project. All rice accessions were cultivated in the same geographical location in Huai'an (119°0'14″E, 33°38'43″N), Jiangsu province in 2019 and 2020. Each accession was subject to the same field management in 2019 and 2020. To eliminate error results caused by marginal effects, every rice accession was planted in a 5 × 5 block within the 3 m × 3 m square, and five plants of each accession were randomly chosen from the middle of each square as the experimental subjects.

### LTG measurement

The seeds of each rice accession were collected independently in a nylon bag with dense nets to air dry seeds for 2 weeks. After that, air-dried seeds were placed in the oven at 50 °C for 7 days to break primary dormancy. A total of 100 plumped seeds of each rice accession were extracted and spread on a round wet filter paper and kept at 15 °C and in darkness for germination. The number of germinated seeds was recorded daily from D6 to D10 with a seed shoot or root exceeding 0.1 cm considered a germinated seed (*Wang et al., 2018c*; *Akhtamov et al., 2020*; *Najeeb et al., 2020*). Seed germination rate = germinated seeds/100. LTG was assessed according to the germination rate of each recorded day.

### SNP filter analysis

The genetic variations of 276 rice accessions are available publicly in the 3K-RG database and the information for all SNPs can be downloaded from the website for free (https://snp-seek.irri.org/_download.zul). In this study, the set criteria for selecting high-quality SNPs were based on (1) minor allele frequency (MAF) ≥0.05 and (2) number of accessions with minor alleles ≥6 (*Yang et al., 2014*). After filtering, only high-quality SNPs were retained. A slide window of 1 Mb was adopted to demonstrate the distribution of variants in all 12 chromosomes to determine the density of the SNPs. The detected SNPs were annotated and the possible effects were predicted through ANNOVAR (*Wang, Li & Hakonarson, 2010*).

## Population structure analysis

To analyze the population structure, a principal component analysis (PCA), a neighbor-joining (NJ) tree and a *K* value analysis were applied. The phylogenetic tree was constructed using MEGA7 (version 7.0) (*Kumar, Stecher & Tamura, 2016*) and the results were visualized using ggtree (version 1.7.10) (*Yu et al., 2017*). The PCA was conducted by PLINK (version v1.90) (*Purcell et al., 2007*). According to the Bayesian Markov Chain Monte Carlo (MCMC) Program, the *K* value, ranging from 2 to 7 in the full population was inferred using STRUCTURE (version 2.3.4) (*Pritchard, Stephens & Donnelly, 2000*). The optimal *K* value was determined by Δ*K* (*Evanno, Regnaut & Goudet, 2005*). The result was visualized and the relevant Q matrix was generated for further analysis.

## Programs for GWAS analysis

Based on the factored spectrally transformed linear mixed model, two programs, FaST-LMM (version 0.5.1) and GEMMA (version 0.98.1), adding different genetic similarities to analyze random effects, were applied to perform the GWAS. The validated number of SNP markers (N) was calculated using the Genetic type I Error Calculator (GEC) software (*Turner, 2014*) and suggestive (1/N) *P* value threshold was adopted as the standard to control type I error.

## Quantitative real-time PCR assay

Ten seeds of each accession were sampled at 15 °C and in darkness. Total RNA was extracted using the TIANGEN RNAprep Pure kit (#DP441; TIANGEN, Beijing, China) according to the manufacturer's protocol. Complementary DNA (cDNA) was synthesized using a cDNA synthesis kit (#RR047A; Takara, Tokyo, Japan). Quantitative real-time PCR (qRT-PCR) reaction was conducted using TB Green Premix Ex Taq (#RR820A; Takara, Tokyo, Japan). The reaction was performed on the CFX Connected Real Time System (Bio-Rad, Hercules, CA, USA). The expression level was calculated by $2^{-\Delta Ct}$ using the expression level of *Ubiquitin* as reference. Each sample was tested three times to fulfill technical replications. Relevant primer sequences are provided in the supplemental data (Table S1).

# RESULTS

## Phenotypic variation for LTG in natural rice accessions

A collection of 276 rice germplasms was selected from the 3K-RG project for the LTG test. Rice accessions in this study were from 17 different regions worldwide (Fig. S1, Table S2). Previous studies have applied different temperatures ranging from 12 °C to 15 °C to estimate LTG (*Borjas, De Leon & Subudhi, 2015*; *Fujino et al., 2008*; *Li et al., 2013*; *Wang et al., 2011*). Given the effect of secondary dormancy induced in 12 °C (*Miura & Araki, 1996*), 15 °C was applied to evaluate LTG in this study. Germination was defined as the seed shoot or root exceeding 0.1 cm from the seed coat (Fig. 1A), and the evaluation of LTG was based on the germination rate from D6 to D10. The average germination rate on 2019D6 was lower (3.5%) than that (13.6%) on 2020D6 (Fig. 1B, Table 1), suggesting that environmental factors have an impact on the phenotype of LTG. Furthermore, the

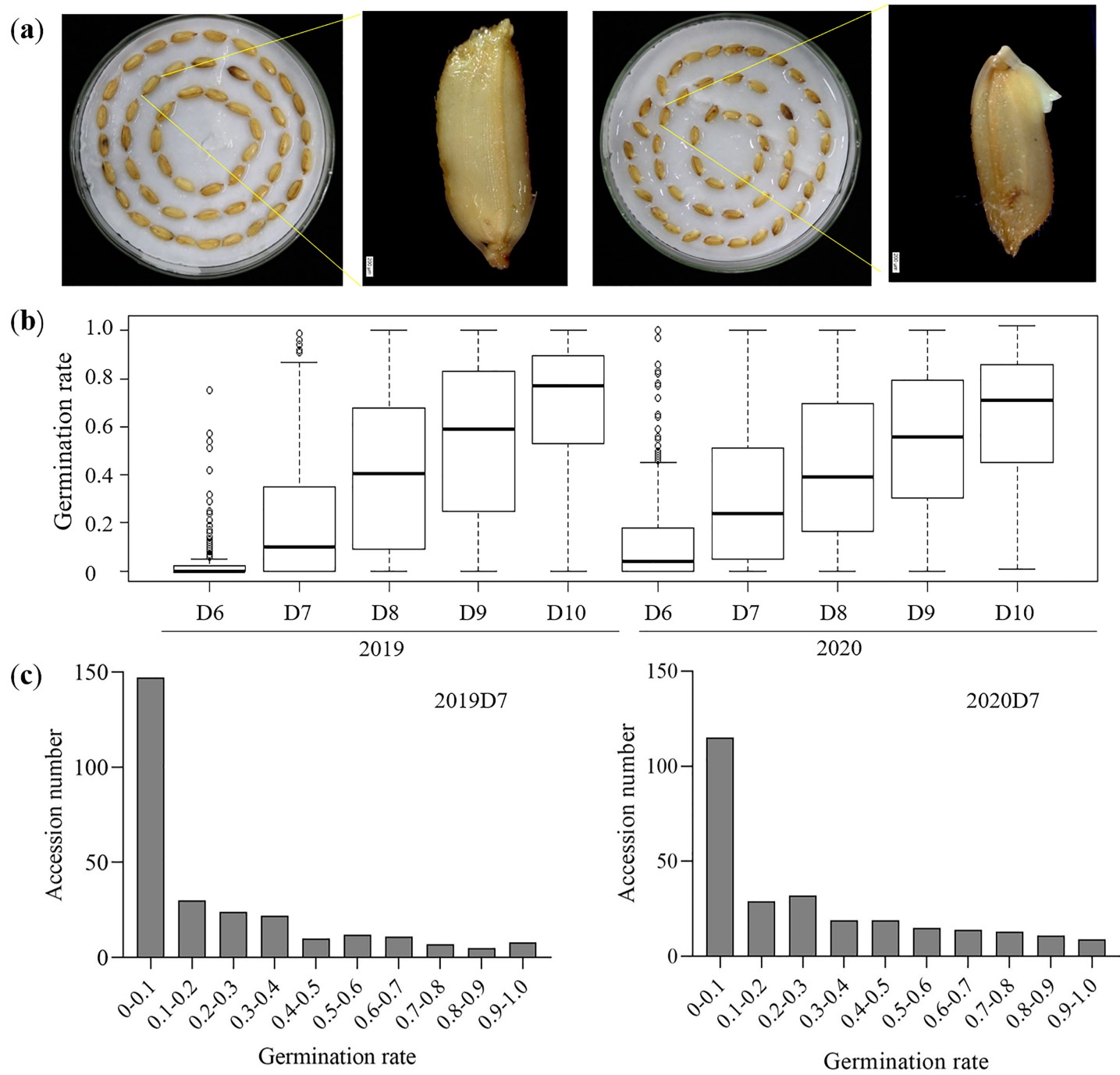

**Figure 1 Description of LTG.** (A) Variations of low temperature germination in D7. Bar = 200 µm. (B) Germination rate from D6 to D10 for two different years. (C) Germination rate distribution on D7 in 2019 and 2020.

total number of germinated accessions was too small to draw any general conclusions (Fig. S2). The average germination rates were 22% and 31% on 2019D7 and 2020D7, respectively (Fig. 1B, Table S3). On 2019D7, a total of 129 rice accessions had a germination rate that exceeded 10% while on 2020D7 this number rose to 161 (Fig. 1C). From D8 to D10 in both years, the average germination rate almost reached or exceeded

**Table 1 Description of germination rate in full population.**

| Days | Germination rate in 2019 | | | Germination rate in 2020 | | |
|------|-------|-----------|--------|-------|-----------|--------|
| | Range | Mean ± SD | Median | Range | Mean ± SD | Median |
| D6 | 0–0.75 | 0.035 ± 0.100 | 0 | 0–1 | 0.136 ± 0.204 | 0.04 |
| D7 | 0–0.99 | 0.220 ± 0.265 | 0.1 | 0–1 | 0.310 ± 0.285 | 0.24 |
| D8 | 0–1 | 0.399 ± 0.315 | 0.405 | 0–1 | 0.429 ± 0.302 | 0.39 |
| D9 | 0–1 | 0.537 ± 0.311 | 0.59 | 0–1 | 0.544 ± 0.286 | 0.56 |
| D10 | 0–1 | 0.694 ± 0.258 | 0.77 | 0.01–1 | 0.647 ± 0.259 | 0.71 |

40% (Fig. 1B, Table 1). The investigation of germination rate gaps between two adjacent days indicated that the gap between D6 and D7 in both years was the largest at 18.5% in 2019 and 17.4% in 2020, respectively (Table 1). In contrast, the smallest germination gap in 2019 was 13.8% between D8 and D9, while in 2020, the gap between D9 and D10 was the smallest (10.3%) (Table 1). On D10 of both years, all rice accessions had a relatively high germination rate ranging from 64.7% to 69.4%. Overall, the distribution of germination rate followed similar trends in both years although the germination rate of D6 and D7 in 2020 was higher than that in 2019 (Fig. 1B, Table 1).

## SNP density analysis

The original version of the 3K-RG database contained 32 million SNPs in total. Through filtering, a total of 497,231 SNPs were detected. After the classification of SNPs, the density of SNPs in all 12 chromosomes were between 1,033.3/1 Mb and 1,648.94/1 Mb (Fig. S3, Table S4). This indicated that the filtered SNPs in this study were sufficient and distributed evenly in 12 chromosomes.

## Population structure and kinship

Using the SNPs, we performed a PCA to quantify the population structure of these 276 accessions. The total variance explained by PC1 and PC2 was 35.60% and 16.79%, respectively (Fig. 2A). Based on the Nei's genetic distance (Nei, 1972), the NJ tree was plotted separating the full group into two groups (Fig. 2B). Meanwhile, using STRUCTURE, the peak of $\Delta K$ appeared when $K = 2$, suggesting that the full population could be divided into two subgroups (Figs. 2C and 2D). These two subgroups corresponded to Japonica and Indica (Table S2), which is consistent with the findings of Wang et al. (2018a).

## GWAS for LTG in rice

A total of 136,276 validated SNPs (MAF ≥ 0.05) were used for the GWAS through the FaST-LMM and GEMMA models. The GEC was used to calculate the indicator $P$ value, which gave 7.41E−6 as the suggestive $P$ value (−log ($p$ value) = 5.13). According to a previous study, the distance of two adjacent lead SNPs within 200 kb was considered one QTL (Lv et al., 2016). A total of 37 QTLs with 54 SNPs were found using FaST-LMM for LTG from D6 to D10 in both years whereas 107 QTLs with 159 SNPs were detected using GEMMA (Table 2, Tables S5–S7). Nearly half of the QTLs identified in FaST-LMM

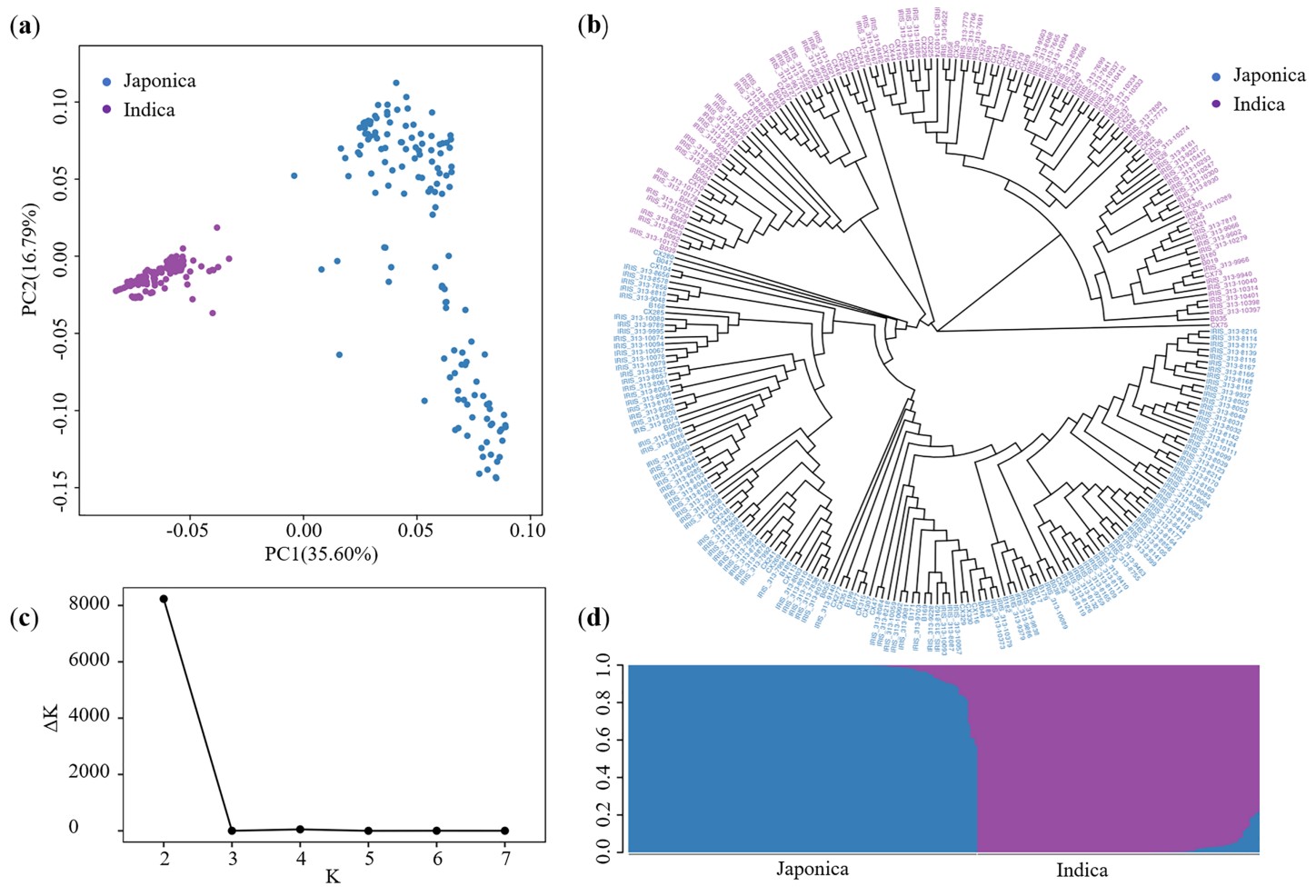

**Figure 2 Description of population structure.** (A) Principal component analysis. (B) NJ tree based on Nei's genetic distance. (C) Delta *K* values plotted as the number of subgroups. (D) Subgroups inferred using STRUCTURE.

(15/37) were also identified in GEMMA, suggesting FaST-LMM had stricter criteria in controlling false positive association (Table S6).

Using FaST-LMM, there were 26, 8, 5, 4 and 4 QTLs detected from D6 to D10, respectively, in 2019 and 2020 (Table 2). There were seven QTLs characterized repeatedly in the total (Table 2). Compared with QTLs reported before, 13 QTLs were co-localized within the interval of 1 Mb in this study (Table 2). Among these QTLs, 12 QTLs were associated with LTG, five QTLs were associated with cold tolerance at the seedling stage, and four QTLs were associated with both LTG and cold tolerance at the seedling stage (Table 2). These results confirmed that the GWAS results in this study were reliable for further candidate analysis. The remaining 24 QTLs that had been uncharacterized before were considered novel QTLs for LTG. Among the novel QTLs, it was notable that *qLTG1-2* was repeatedly detected on D7 in both years using FaST-LMM and GEMMA (Figs. 3A–3C). Moreover, this QTL was detected on 2020D6 as well using FaST-LMM and GEMMA (Fig. S4). Therefore, further analysis was focused on *qLTG1-2* with this repeated lead SNP (Chr.1_13340259).

**Table 2 Summary of detected QTLs using FaST-LMM in the full population.**

| QTLs | Trait ID | Chromosome | Peak SNP | *p*-value | Reported QTLs overlapped |
|---|---|---|---|---|---|
| *qLTG1-1* | 2019D6 | Chr1 | 12,153,951 | 5.51 | *qCTGERM1-5 (Shakiba et al., 2017)* |
| *qLTG1-2* | 2019D6, 2019D7, 2020D6, 2020D7 | Chr1 | 13,340,259 | 5.83 | |
| *qLTG1-3* | 2020D6 | Chr1 | 19,239,470 | 6.09 | *qCTS1-2 (Wang et al., 2016)* |
| *qLTG1-4* | 2019D6 | Chr1 | 22,886,860 | 6.85 | *qCTGERM1-6 (Shakiba et al., 2017)* |
| *qLTG1-5* | 2020D6 | Chr1 | 24,833,598 | 5.60 | |
| *qLTG1-6* | 2020D10 | Chr1 | 29,923,602 | 5.88 | |
| *qLTG1-7* | 2019D6 | Chr1 | 35,250,579 | 5.23 | *qLTG1b (Fujino et al., 2015)* |
| *qLTG2-1* | 2019D6 | Chr2 | 4,583,247 | 5.73 | *OsWRKY71, qCTS2-2* and *qLTGS(III)2 (Kim et al., 2016; Wang et al., 2016; Najeeb et al., 2020)* |
| *qLTG2-2* | 2019D6 | Chr2 | 20,749,806 | 6.10 | *qLTG(III)2 (Najeeb et al., 2020)* |
| *qLTG2-3* | 2020D6, 2020D7, 2020D8 | Chr2 | 26,062,949 | 5.54 | |
| *qLTG2-4* | 2019D6 | Chr2 | 30,309,540 | 7.91 | *OsMADS57 (Guo et al., 2013)* |
| *qLTG2-5* | 2019D6 | Chr2 | 30,974,975 | 8.54 | |
| *qLTG3-1* | 2019D6 | Chr3 | 24,070,502 | 6.09 | |
| *qLTG4-1* | 2020D10 | Chr4 | 2,756,738 | 5.28 | *qCTGERM4-3 (Shakiba et al., 2017)* |
| *qLTG4-2* | 2020D8, 2020D9 | Chr4 | 3,566,435 | 5.35 | |
| *qLTG4-3* | 2020D8, 2020D9, 2020D10 | Chr4 | 4,192,136 | 6.74 | |
| *qLTG4-4* | 2020D6, 2020D7 | Chr4 | 4,527,433 | 5.14 | *qLTG(II)4–2 (Najeeb et al., 2020)* |
| *qLTG4-5* | 2019D6 | Chr4 | 20,867,550 | 5.20 | |
| *qLTG4-6* | 2019D6 | Chr4 | 23,131,460 | 5.27 | |
| *qLTG6-1* | 2019D6 | Chr6 | 20,322,237 | 6.07 | |
| *qLTG7-1* | 2020D6 | Chr7 | 1,702,699 | 6.14 | *qLTG7* and *qCTS7-1 (Fujino et al., 2015; Wang et al., 2016)* |
| *qLTG7-2* | 2020D7 | Chr7 | 5,701,029 | 5.25 | |
| *qLTG7-3* | 2020D6 | Chr7 | 11,338,200 | 5.47 | |
| *qLTG7-4* | 2020D8 | Chr7 | 13,267,244 | 5.21 | *qCTGERM7-2 (Shakiba et al., 2017)* |
| *qLTG7-5* | 2020D6 | Chr7 | 14,587,580 | 6.83 | |
| *qLTG7-6* | 2019D6 | Chr7 | 28,676,190 | 6.33 | *qCTS7-5* and *qCTGERM7-5 (Wang et al., 2016; Shakiba et al., 2017)* |
| *qLTG8-1* | 2020D6 | Chr8 | 6,167,751 | 5.66 | |
| *qLTG8-2* | 2020D6 | Chr8 | 7,601,891 | 5.80 | |
| *qLTG9-1* | 2019D6 | Chr9 | 7,410,218 | 6.22 | |
| *qLTG10-1* | 2019D6 | Chr10 | 23,066,742 | 6.61 | *qCTGERM10-4 (Shakiba et al., 2017)* |
| *qLTG11-1* | 2020D7, 2020D8 | Chr11 | 1,170,653 | 5.58 | |
| *qLTG11-2* | 2019D6 | Chr11 | 17,712,316 | 5.55 | *qCTS11-5* and *qCTGERM11-4 (Wang et al., 2016; Shakiba et al., 2017)* |
| *qLTG12-1* | 2019D6 | Chr12 | 1,512,598 | 5.40 | |
| *qLTG12-2* | 2020D7, 2020D9, 2020D10 | Chr12 | 2,084,623 | 5.14 | |
| *qLTG12-3* | 2019D7 | Chr12 | 10,140,027 | 5.47 | |
| *qLTG12-4* | 2019D7 | Chr12 | 11,182,503 | 5.93 | |
| *qLTG12-5* | 2019D9 | Chr12 | 23,640,519 | 5.56 | |

 

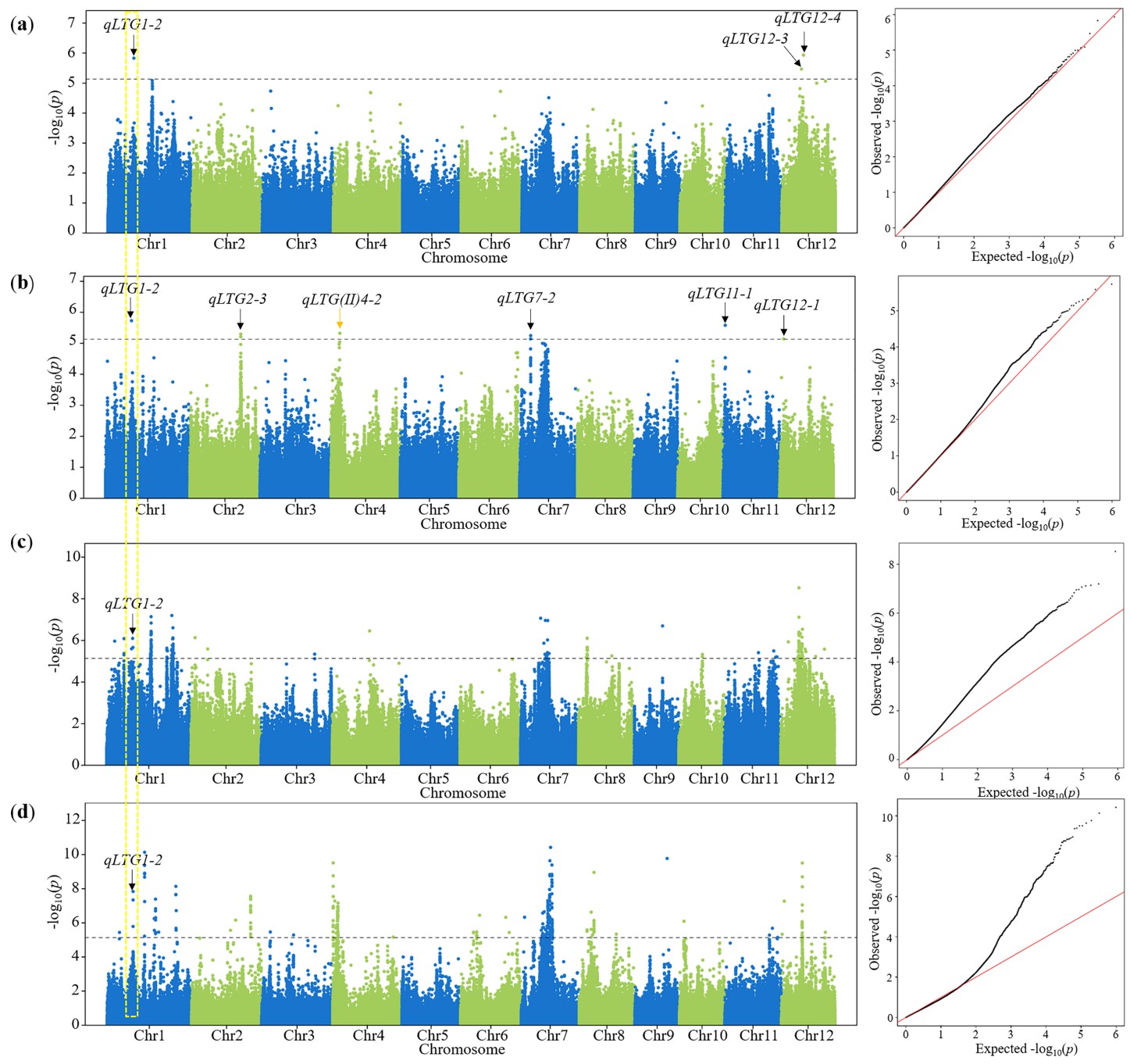

**Figure 3 Manhattan plot and Q–Q plot for LTG using 2 programs in D7.** (A) A GWAS performed on 2019D7 using FaST-LMM. (B) GWAS performed on 2020D7 using FaSTL-MM. (C) A GWAS performed on 2019D7 using GEMMA. (D) A GWAS performed on 2020D7 using GEMMA. An orange arrow represents QTLs detected previously. A black arrow represents novel QTLs detected in this study. A yellow dotted box represents the repeated identified QTLs. A dashed horizontal line represents the suggestive threshold ($P = 7.34 \times 10^{-6}$, $-\log_{10} P = 5.13$).

## Haplotype and expression analysis of the candidate genes

To further locate the candidate genes of *qLTG1-2*, two public germination-related transcriptome datasets (SRP277875, GSE115371) were adopted. Dataset SRP277875

contained the expression data at different time points of two rice accessions for germination (*Yang et al., 2020a*) and Dataset GSE115371 provided the expression data of one rice accession under aerobic conditions for germination at various time points (*Narsai et al., 2017*). According to the published transcriptome data, 13 expressed genes located in the *qLTG1-2* interval were found, which were then used for further comparison analysis of the expression levels in low temperature between high and low germination accessions (Fig. S5). *LOC_Os01g23600*, *LOC_Os01g23705* and *LOC_Os01g23850* failed to be amplified, suggesting they exhibit very low expression levels in the seeds. Eight genes, including *LOC_Os01g23590*, *LOC_Os01g23610*, *LOC_Os01g23630*, *LOC_Os01g23640*, *LOC_Os01g23680*, *LOC_Os01g23710*, *LOC_Os01g23740* and *LOC_Os01g23870*, did not show obvious differences between high and low germination accessions (Fig. S6).

    *LOC_Os01g23580* was located 90 kb from the lead SNP and was associated with abiotic stress in a GO analysis. Furthermore, the homolog of *LOC_Os01g23580* in *Arabidopsis* has been shown to be involved in the regulation of auxin transport and response to several abiotic stresses (*Li et al., 2005*; *Wijewardene et al., 2020*). One non-synonymous SNP (Chr.1_13243045, base G-C, amino acid Ser-Thr) and one upstream SNP (Chr.1_13236390, base A-G) were identified within the sequence of *LOC_Os01g23580*, which generated three haplotypes in the full population (Fig. 4A). Haplotype I of *LOC_Os01g23580* displayed better performance for LTG than the left two haplotypes (Figs. 4A and 4B). Moreover, accessions of high germination rates were usually ones with G allele whose transcriptional levels were much lower than accessions of low germination rates with an A allele in the upstream region (Fig. 4C). *LOC_Os01g23620*, namely *OsSar1a*, was located 50 kb from the lead SNP and *OsSar1abc* RNAi mutants led to pre-harvest sprouting (*Tian et al., 2013*). Based on one upstream SNP (Chr.1_13285882 base A-G), the full population was divided into two haplotypes (Fig. 4D). Haplotype I of *OsSar1a* showed a higher germination rate than haplotype II which was negatively associated with transcriptional level (Figs. 4E and 4F).

## DISCUSSION

The genetic variation of rice cultivars provides a resource for trait improvement *via* breeding (*Breseghello & Coelho, 2013*). The 3K-RG project provides a foundation for finding potential candidate genes associated with quantitative traits (*Wang et al., 2018a*). Using rice accessions from the 3K-RG project, several genes for crucial agronomic traits were identified (*Anacleto et al., 2019*; *Kumar et al., 2020*; *Lu et al., 2021*).

    LTG is an essential agronomic trait for direct seeding rice in high altitude regions (*Li et al., 2013*). In previous studies, LTG was measured using two parameters: low temperature germination index (LTGI) and low temperature germination potential (LTGP) (*Ji et al., 2009*; *Wang et al., 2018b*). Since germination varies greatly in natural accessions, LTG was generated according to daily germination rates (*Fujino et al., 2004*). Although accessions in both years of this study had similar patterns of germination rates, a few of them differed in the early days of germination, indicating that environmental factors could not be ignored (Fig. 1B).
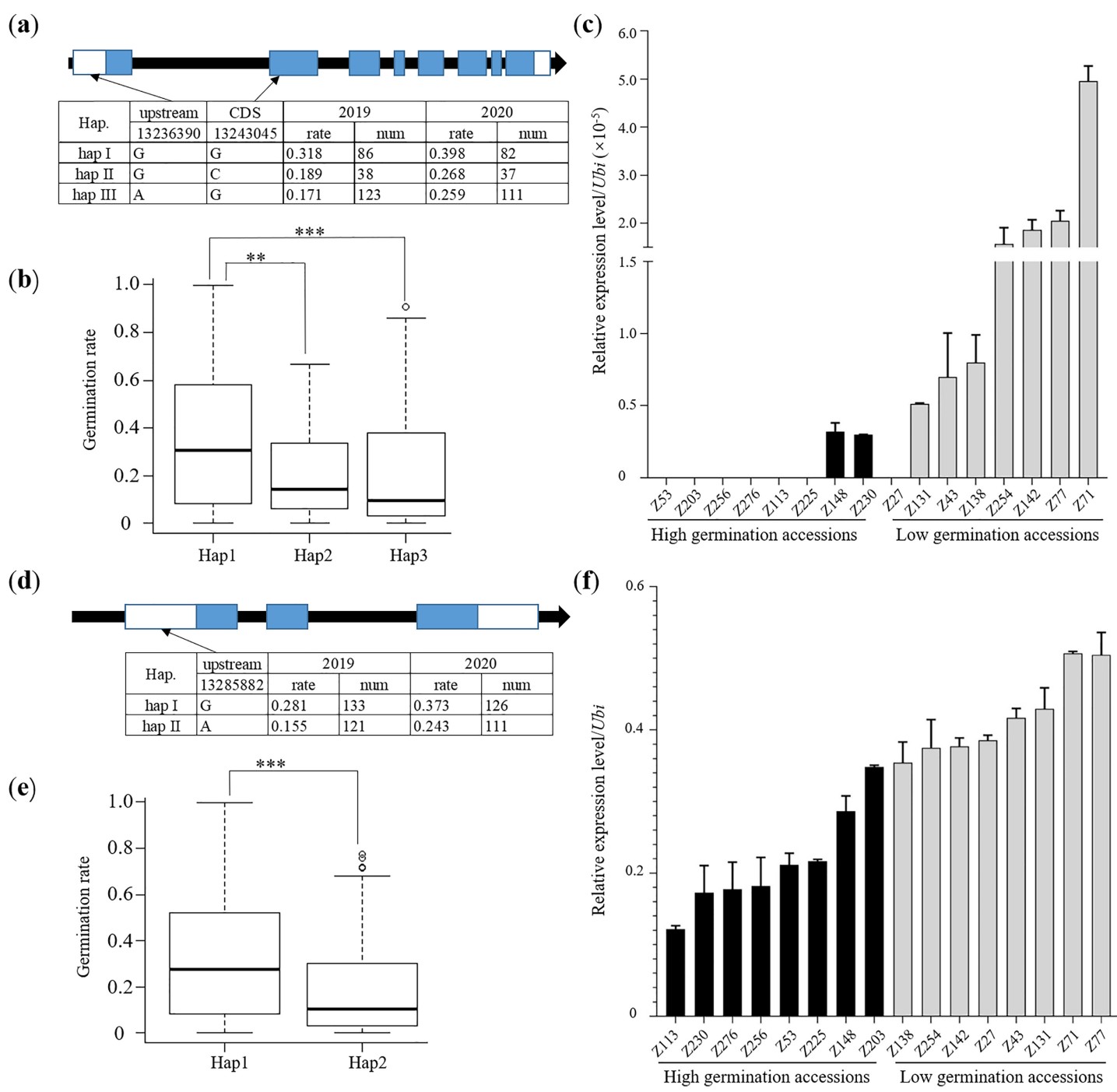

**Figure 4 Candidate genes analysis.** (A) Gene structure and haplotype analysis for *LOC_Os01g23580*. (B) Comparison of germination rate among *LOC_Os01g23580* haplotypes in full population (**$p < 0.01$; ***$p < 0.001$). (C) Expression level of *LOC_Os01g23580* in contrast accessions after 3 days soaking in water in 15 °C and darkness for germination. Black bars represented expression levels of rice accessions with high germination rate under low temperature. Grey bars represent the expression levels of rice accessions with low germination rate under low temperature. (D) Gene structure and haplotype analysis for *LOC_Os01g23620*. (E) Comparison of germination rate among *LOC_Os01g23620* haplotypes in full population (**$p < 0.01$). (F) Expression level of *LOC_Os01g23620* in contrast accessions after 3 days soaking in water in 15 °C and darkness for germination.

In both study years, four accessions, 'IRIS_313-7728', 'B199', 'B077' and 'IRIS_313-9886' showed an extremely high germination rate on D6 in low temperatures. Thus, these four accessions could be considered potential donors for rice breeding with regard to LTG.

A GWAS was also performed in Japonica and Indica subgroups, separately, using FaST-LMM. A total of 21 and 33 QTLs with 49 and 37 SNPs were mapped in Japonica and Indica, respectively, for LTG in both years (Tables S8–S11). For both subgroups, the GWAS results for LTG were consistent with those in the full population. In the Japonica group, 11 QTLs overlapped within the interval of QTLs mapped previously (*Fujino et al., 2015*; *Najeeb et al., 2020*; *Shakiba et al., 2017*; *Wang et al., 2016*) (Table 2, Table S10), of which four QTLs were also detected in the full population. Coincidentally, in the Indica group, there were also 11 QTLs that had been mapped previously and four of them were also found in the full population (Table 2, Table S11) (*Fujino et al., 2015*; *Najeeb et al., 2020*; *Shakiba et al., 2017*; *Wang et al., 2016*). These analyses confirmed the GWAS results in the full population in this study.

The repeatedly detected QTL (*qLTG1-2*) in the full group was also found in the Japonica (*qLTG-1-1-2*) and Indica (*qLTG-2-1-1*) subgroups (Table 2, Tables S10 and S11). Two candidate genes showed different expression levels in contrast with germination rate varieties (Figs. 4C and 4F). *OsSar1a* (*LOC_Os01g23620*) was functionally identified to be involved in seed germination (*Tian et al., 2013*). According to the haplotype analysis, accessions with a G allele variant located within the 1 kb upstream region of *OsSar1a* showed higher germination rates (Table S2). Through a *cis*-element analysis (http://bioinformatics.psb.ugent.be/webtools/plantcare/html/), accessions with A allele had complete CAAT-box functions as an enhancer motif in the promoter region (Fig. 4D, Fig. S7). In agreement with these results, the SNP variant A is associated with high transcriptional levels (Fig. 4F). These results indicate *OsSar1a* could be a promising candidate gene for LTG in rice breeding. So far, few reports have clarified the function of *LOC_Os01g23580*, but its homolog in *Arabidopsis* is involved in the regulation of auxin transport and confers tolerance to various stresses (*Li et al., 2005*). Further elucidating the biological function of *LOC_Os01g23580* may be important for rice breeding application.

## CONCLUSION

A set of 276 rice accessions from the 3K-RG project with 497 k re-sequenced SNPs was used for a GWAS to uncover candidate genes regulating LTG. Combined with the phenotypes from two consecutive years, a total of 37 QTLs were identified in the full population, co-localizing with 12 QTLs reported before for LTG. Among all QTLs, one novel QTL, *qLTG1-2* was detected repeatedly in both study years by both the FaST-LMM and GEMMA programs. Based on two published transcriptome datasets, a total of 13 seed-expressed genes were identified for a haplotype analysis and expression analysis. Eventually, two promising candidate genes, *OsSar1a* (*LOC_Os01g23620*) and *LOC_Os01g23580*, which both showed differential expression levels in the accessions of contrast LTG traits, were explored as favorable haplotypes of LTG for rice direct seeding.

These results may be helpful for further developing rice varieties with high LTG for rice direct seeding through marker-assisted breeding.

### Funding

This work was supported by the Innovation Program Foundation of Colleges of Jiangsu Province, China (202110323059Y) and the Natural Science Foundation of Colleges of Jiangsu Province, China (20KJB210003). The funders had no role in study design, data collection and analysis, decision to publish, or preparation of the manuscript.

### Grant Disclosures

The following grant information was disclosed by the authors:
Innovation Program Foundation of Colleges of Jiangsu Province, China: 202110323059Y.
Natural Science Foundation of Colleges of Jiangsu Province, China: 20KJB210003.

### Competing Interests

The authors declare that they have no competing interests.

### Author Contributions

- Feng Mao performed the experiments, analyzed the data, prepared figures and/or tables, authored or reviewed drafts of the paper, and approved the final draft.
- Depeng Wu analyzed the data, authored or reviewed drafts of the paper, and approved the final draft.
- Fangfang Lu performed the experiments, analyzed the data, prepared figures and/or tables, and approved the final draft.
- Xin Yi performed the experiments, analyzed the data, prepared figures and/or tables, and approved the final draft.
- Yujuan Gu analyzed the data, prepared figures and/or tables, and approved the final draft.
- Bin Liu analyzed the data, prepared figures and/or tables, and approved the final draft.
- Fuxia Liu performed the experiments, analyzed the data, prepared figures and/or tables, and approved the final draft.
- Tang Tang performed the experiments, analyzed the data, prepared figures and/or tables, and approved the final draft.
- Jianxin Shi conceived and designed the experiments, authored or reviewed drafts of the paper, and approved the final draft.
- Xiangxiang Zhao conceived and designed the experiments, authored or reviewed drafts of the paper, and approved the final draft.
- Lei Liu conceived and designed the experiments, performed the experiments, analyzed the data, prepared figures and/or tables, authored or reviewed drafts of the paper, and approved the final draft.
- Lilian Ji conceived and designed the experiments, authored or reviewed drafts of the paper, and approved the final draft.
## Data Availability

The raw data is available as a Supplemental File.

## Supplemental Information

Supplemental information for this article can be found online at http://dx.doi.org/10.7717/peerj.13407#supplemental-information.

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
