# Peer review of "QTL mapping and candidate gene analysis of low temperature germination in rice (Oryza sativa L.) using a genome wide association study"

_PeerJ, doi:10.7717/peerj.13407_

## Round 0.1 · original submission · Major Revisions

Both reviewers suggested a number of improvements to your manuscript. In particular, you should explain the relevance and mechanisms of low-temperature germination.

Reviewer 2 made several comments you should address as well.

1

Reviewer 1 ·

Basic reporting

The thesis is well organized and described logically.

But, It is necessary to review recent papers related to low-temperature germination.
Moreover, there are many articles for GWAS related to low-temperature germination,

There are some errors in some references.
line 160; not in the reference.
line 315 & line 317: need to be moved over line 337.

Experimental design

In GWAS, the plant population is very important.
Authors need to describe how to select a mini core collection (276) from 3K-RG.

Validity of the findings

The authors need to add the results of the GWAS using GEMMA analysis for 2020D7 in Figure 3.
Please describe how to select two genes out of 13 genes.

Need to modify.
4.1% in line 129 and 22% in line 133 should be modified to 3.5% and 24% according to Table 1, respectively.

Additional comments

In fig 4. please check the variety accessions. The names differ in footnote and picture.

Reviewer 2 ·

Basic reporting

- The authors should devote a paragraph or so introducing LTG and why it is an important agronomic trait. - The introduction should answer quite obvious questions that come to mind for someone not familiar with this trait, for example: what is its biology? Why should I care for this trait? Why do the authors care for this trait? Why did you use this approach to dissect the trait architecture? What other approaches could you have used-why didn’t you use those approaches? etc.
- L 37: What makes rice the MOST IMPORTANT staple? Provide more recent and relevant support for your statement, otherwise, revise accordingly.
- It makes little sense to write the Introduction in the past tense – Lines: 39,41,52,53, etc
- Present a coherent discussion of the studies you list in the introduction – why are they relevant to the reader?

Experimental design

- L 71 / L 122: Be consistent - is this a mini core collection? Cite or provide a description of the collection if it is.
- L 73: “… Meanwhile each accession was treated equally as much as possible in two consecutive years …” what do you mean by this?
- L 128: “… Germination is defined as the seed shoot or root exceeding 0.1 cm from the seed coat …” Why 0.1 cm? Provide relevant support.
L 129: Justify the choice of time points: why D6 to D10 – why not D5 or D4 to D10 or more?
L 130: Why could this have been the case – do you have any speculations?
- How did you use the transcriptome datasets for haplotype analysis?

Validity of the findings

- Can you clarify what the percentages mean? For example, L 133 – 134: “… average germination rates were 22% and 31% with a medium of 10% and 22% in 2019 and 2020, respectively …” What does the medium statistic represent?
- L 150 – 156: This is an interesting result. How much variation do PC 1 and 2 account for together? Is this the correspondence between subgroup membership 100% consistent with Japonica and Indica – how do you explain the discrepancy if any?

- L 158: “… A total of 136,276 validated SNPs were calculated in silico. …” what do you mean by calculated in silico?
- L 159: “… Based on the calculation of P value, 5.13 was set as the threshold in this study for the full population. …” What P value calculation are you referring to??

Additional comments

L 150 - 156: Since this is quite an interesting result ( the correspondence between these ~270 accessions with rice sub-classification Japonica and Indica, it would make sense to include in the introduction brief material about the botany of rice.

Annotated reviews are not available for download in order to protect the identity of reviewers who chose to remain anonymous.

---

## Round 0.2 · Minor Revisions

Although your paper is scientifically sound, there are numerous grammar errors diminishing the value of your work. Thus, we would urge you to contract a professional proofreading service for making your manuscript publishable.

Reviewer 2 ·

Basic reporting

The manuscript could benefit from professional English translation. Grammatical errors are a major distraction from an otherwise good paper.

The authors responded well to suggestions for more background with a sufficient review of relevant literature on the subject of study.

Experimental design

Sound and valid experimental design were used.

Validity of the findings

Valid but not novel.

---

## Round 0.3 · accepted · Accept

Thank you very much for your contribution and for polishing your manuscript.